# Air Quality, Management Practices and Calf Health in Italian Dairy Cattle Farms

**DOI:** 10.3390/ani12172286

**Published:** 2022-09-03

**Authors:** Serena Bonizzi, Giulia Gislon, Milena Brasca, Stefano Morandi, Anna Sandrucci, Maddalena Zucali

**Affiliations:** 1Department of Agricultural and Environmental Sciences, University of Milan, Via Celoria 2, 20133 Milan, Italy; 2Italian National Research Council, Institute of Sciences of Food Production ISPA, Via Celoria 2, 20133 Milan, Italy

**Keywords:** dairy farm management, air quality, microbial contamination, calf health

## Abstract

**Simple Summary:**

Calf mortality is an important factor of economic loss in dairy operations. Among the factors which can affect calf health, air quality in calf housing has rarely been studied. In the present research, the relations among air quality in the calf pens, management practices, microclimatic conditions and calf health and mortality were studied in 25 Italian dairy cattle farms. Microbial air quality in the calf pens was found to be related to temperature and humidity, design and position of the pen, some management practices and calf health and mortality. Maintaining dry bedding and controlling microclimatic conditions in the calf pen can be useful practices to promote good air microbiological quality in the calf pen, improving calf health and reducing mortality.

**Abstract:**

Among factors that can affect calf health, microbial quality of the pen air is poorly studied. In 25 Italian dairy farms, data concerning air quality in the calf pens, hygiene of pens and equipment, microclimatic conditions, calf health and management were collected during the winter season (January-March 2020 and December-March 2021). The average air Standard Plate Count (SPC) of 85 pens was 4.51 (SD = 0.52) log_10_ cfu/m^3^ whereas the average air ammonia concentration was 0.66 (SD = 0.53) ppm. Positive correlations were found between average Temperature Humidity Index (THI) in the pen and air SPC, night maximum THI and air SPC and between SPC and yeast, mould and ammonia concentration in the pen air. The concentrations of *E. coli*, Enterobacteriaceae and yeasts in the pen air were higher and calf cough increased as the renewal interval of bedding material became longer. High bedding dry matter and low THI were associated with low air SPC, good calf health scores and low mortality. Maintaining low bedding humidity and controlling microclimatic conditions can contribute to enhancing air microbiological quality in the pen and reduce calf diseases and mortality.

## 1. Introduction

The success of dairy cattle husbandry is closely linked to good management of young animals and in particular of the female calves, which represent the future of the dairy herd. Calf mortality is an important cost item in dairy operations due to the loss of the present value of the calf and the loss of genetic potential for herd improvement. The main causes of calf death during the first month of life are gastrointestinal disorders and pneumonia; they are mainly influenced by housing conditions, colostrum intake and feeding management [1].

Different housing systems have been developed to protect calves from extreme climatic conditions and thermic stress. Depending on the climate, culture, and intended use of animals, these include open range, shelters, pens, hutches, and naturally or mechanically ventilated barns [2]. Housing design is an important factor affecting humidity and the proliferation of microorganisms in the barns [3]. Bedding also plays an important role in calf management: studies [4] reported that dairy calves typically spend about 18 h/d lying down and data suggest that inadequate lying times reduce growth. In particular, young animals show a preference for dry sawdust-bedded lying surfaces and avoid lying on wet bedding. The quality of bedding material affects the heat loss by conduction in lying calves. In particular, deep straw bedding is superior to other bedding materials as an insulator and can provide a high ‘nesting score’ which has a preventive effect against respiratory diseases in calves [5].

Although some housing systems succeed in providing thermal comfort to calves, other problems, such as poor air quality, might arise [2]. Unfortunately, very little is known about the effect of air quality in calf housing. The indoor air of dairy cattle barns usually harbours variable concentrations of microorganisms, depending on different factors, such as animal density, management, technologies, floor and bedding materials, microclimate conditions and, especially, ventilation rate [6]. Temperature and humidity in the barn influence environmental contamination as well as animal welfare [3]. According to other authors [6], concentration and quality of airborne microbiota in the air of dairy farms affects the health of animals and human workers and not lastly the degree of bacterial pollution in the nearby environment. Other authors [7] showed that 5000 to 30,000 cfu/m^3^ are expected in well-ventilated barns. Bacterial counts exceeding 100,000 cfu/m^3^ in the case of poorly ventilated barns are associated with calf pneumonia [6,7]. Others [8], referring exclusively to calves, stated that a total Bacterial Air Count less than 15,000 cfu/m^3^ lowers the risk of lung problems. Among the most important calf diseases which can be related to pen crowdedness and non-hygienic environment bovine respiratory diseases (BRD) [9] and infectious diarrhoea [10] play a major role. However, there are very few specific indications on the microbial quality of air in calf housing. The concentration of ammonia inside the calf pen is very important as well. Calves are likely to be more sensitive to environmental irritants, such as ammonia, than adult cattle [11]. Ammonia concentration depends on the accumulation of urine and faeces, which emphasises the need for regular cleaning and provision of dry bedding, together with adequate ventilation. According to other authors [5], ammonia levels of less than 10 ppm are recommended. Studies [11,12] suggested ammonia concentration in calf housing lower than 3.5–7.0 ppm and 4.7–6 ppm, respectively. Other authors [13] reported lower risk for respiratory diseases with ammonia concentration below 6 ppm.

Calves are born with functional thermoregulatory mechanisms. Therefore, healthy calves are able to cope with outside temperatures as long as they receive an adequate amount of energy through feed and are provided with a dry housing and deep litter and free from drafts. The lower critical temperature, at which additional energy is needed for heat production, lies in the range of 10–15 °C for calves in the first two weeks of life, declining with age to approximately 6–10 °C in older calves, and is highly dependent on air speed [5].

The main objective of the present work was to study the air microbiological contamination and ammonia concentration in the calf pens, the environmental and management factors affecting air quality and the relationship with calf health and mortality. A secondary aim of the study was to represent the calf management practices adopted in intensive dairy cattle farms in northern Italy.

## 2. Materials and Methods

During the winter season, from January to March 2020 and from December to March 2021, data related to air quality in the calf pens (*Escherichia coli*, Enterobacteriaceae, Standard Plate Count, yeasts, moulds and ammonia concentrations), hygiene of pen walls and feeding equipment, microclimatic conditions and calf health were collected in 25 dairy cattle farms. Moreover, in the same farms a survey was conducted through interviews with the farmers to collect information about pre weaning calf management.

Selection criteria of the farms were based on husbandry techniques, herd size and average daily milk production per head to obtain a sample representative of the intensive dairy cattle husbandry of the Po plain (northern Italy). In all the 25 farms, cows were housed in free-stall barns, without access to pasture, provided with lying cubicles bedded with straw or sawdust. Average individual milk yield was 33.6 (SD = 4.92) kg/day and average herd size was 201 (SD = 153) lactating cows. Most of the farms reared Italian Holstein cows only, whereas 32% of farms had also other cattle breeds (Jersey, Brown Swiss and half-breed cows).

Each farm was visited once to carry out a survey about calf management procedures, to collect samples to be analysed and to perform direct observations. The survey was performed through direct interviews with the farmers; a total of 48 questions were asked regarding calving routine, first care after birth, calf housing, bedding material and management, colostrum and milk feeding. The questionnaire is reported in the Appendix A.

### 2.1. Air Quality Evaluation

Active air sampling was performed, by means of a single-stage sampler with orthogonal impact (Surface Air System^®^ Super ISO, SAS), and Petri dishes containing suitable culture media for microbial counts in particular for *Escherichia coli*, Enterobacteriaceae, Standard Plate Count (SPC), yeast and mould counts. For each Petri dish, a volume of 10 L of air was collected in a few seconds at the level of calf muzzle in at least 3 different pens per farm, depending on the number of pens occupied at the time of visit, for a total amount of 85 pens sampled. As sometimes there were pens of different design in the same farm, pens to be sampled were selected on the basis of design (slatted floor; floor standing; other types including hutches and group pens) and position (central or external, with reference to the position in the line of calf pens next to each other). After sampling, Petri dishes were incubated as soon as possible but always in the same day of collection. Enterobacteriaceae and *Escherichia coli* (the latter with a specific growth substrate) were determined on selective chromogenic medium SENECA EE-EC agar (Biolife Italiana, Milano, Italy) incubated at 37 °C for 24 h. Standard Plate Count was evaluated on Plate Count Agar (Biolife Italiana) incubated at 30 °C for 72 h, and yeasts and moulds on Chloramphenical Glucose Agar (CGA) at 25 °C for 5 days. Following incubation, the numbers of colony forming units of bacteria and fungi in each plate were counted and the obtained values were then converted into the Most Probable Number (MPN/m^3^) using the conversion table provided by the sampler manufacturer. Air microbiological data were analysed as log_10_. 

Ammonia air concentration was measured in the same calf pens selected for air microbial analyses, by using Drager X-am^®^ short-term ammonia tubes (Ammonia 2/a, #6733231) and the Drager^®^ Accuro pump (Drager X-am^®^ Safety AG & Co., Leubeck, Germany). Ammonia concentration was measured on air samples collected placing the instrument at the level of the calf’s muzzle. The ammonia tubes used in this study had a detection range from 1.4 to 21 mg m^−3^. Ammonia concentrations were expressed as parts per million (ppm).

In the present study each farm was visited once, and the air samples were taken in the selected pens regardless of the age of the calves in the pens. In order to test the correctness of the experimental method, in one farm air samples were taken over the course of five weeks in 3 pens to observe possible changes over time. In particular, the samplings were taken in the cages from the day they were empty till five weeks after the entry of the calf.

### 2.2. Hygiene of Calf Pen Walls and Feeding Equipment

Hygienic conditions of pen walls and milk/colostrum feeding equipment were assessed using Bioluminometer System Sure II Plus supplied by Hygiena^®^. This instrument allows us to measure ATP presence on the surface swabbed. The operating instructions of the instrument indicate any clean surfaces with values < 10 Relative Luminescence Units (RLU); the greater the RLU output, the higher the contamination of the surface tested. The walls of pens provided with solid panels (*n* = 43) and the equipment used for colostrum and milk administration (baby bottles, buckets, esophageal probes; *n* = 65) were tested. RLU data were analysed as log_10_.

The Dry Matter (DM) of bedding material of calf pens were also determined pooling bedding from the different pens tested in each farm. Samples were oven dried at 55 °C and repeated weight measurements were carried out at day 2, 3, 7, until stabilisation.

### 2.3. Calf Health Evaluation and Colostrum IgG

The assessment of calf heath status was performed using a protocol derived from the “Calf Health Scoring Chart”, developed by the University of Wisconsin-Madison [14] by 2 observers at the same time. All of the pre-weaning female calves present at the time of the farm visit were evaluated (*n* = 203).

This method was chosen as a quick, simple and non-invasive approach to evaluate calf health status. The protocol was applied only for observable data, excluding rectal temperature measurement and without touching or manipulating calves.

The chart evaluates each indicator assigning a score from 0 to 3, considering 0 as normal status and 3 severely abnormal. The percentage of 0 scores was calculated for each of the following variables: presence of cough, nasal mucus discharge, eye discharge, ear position. As suggested by other authors [14] among eye discharge and ear position scores only the highest one was considered. In the protocol the evaluation of faecal consistency was included as well, assigning 0 for normal consistency and 3 for watery faeces.

Concentration of IgG in colostrum was determined by laboratory analysis, through electrophoresis. Protein fractions were determined by electrophoretic separation using the Hydragel protein kit from Sebia (Issy Les Moulineaux, France), and protein was quantified using a densitometer from CGA (Florence, Italy).

### 2.4. Microclimatic Conditions

Environmental conditions were measured by using HOBO^®^ data loggers supplied by ONSET^®^. The data loggers were anchored to the wall inside the calf pen or placed in a position representative of the microclimatic condition of a group of pens and left for at least 5 days. The devices recorded ambient temperature (AT) in a range from −20 to 70 °C, relative humidity (RH) from 5 to 95% and light intensity from 1 to 4500 lux. Variables were recorded at 5-min intervals, and the Temperature-Humidity Index (THI) was calculated as follows [15]:(1)THI=1.8∗AT−(1−RH)∗(AT−14.3)+32
where:

*AT* = Ambient Temperature (°C)

*RH* = Relative Humidity (%)

### 2.5. Statistical Analysis

The whole dataset, including farm characteristics and management, environmental variables, microbial air contamination, bedding DM, calf health scores, hygiene of pen walls and feeding equipment, was analyzed using SAS software (Version 9.4, 2012). Descriptive statistics were performed, in order to first explore the dataset and check data normality using MEAN, CHART, UNIVARIATE and FREQ procedures. Pearson correlation analysis was performed by using the CORR procedure. Proc GLM was used to evaluate the effect of calf pen position (external, central, with reference to the position in the lines of pens next to each other; Model 1); calf housing design (slatted floor, floor standiI ng, other pen types; Model 2), bedding renewal frequency intended as complete bedding change (<40 days; ≥40 days; Model 2) on air quality characteristics, housing hygiene and calf health scores.

Model 1:(2)$$Y_{ijk}=\mu +P_{i}+F_{j}+e_{ijk}$$
where:

*Y_ijk_* = dependent variables;

*μ* = general mean;

*Pi* = calf pen position effect (i = 1–2; external vs. central);

*Fj* = farm effect (J = 1–25);

*e_ijk_* = residual error.

Model 2:(3)$$Y_{ijkl}=\mu +R_{i\;}or\;T_{i}+D_{j}+e_{ijk}$$
where:

*Yijk* = dependent variables;

*μ* = general mean;

*Ri* = bedding renewal frequency effect (i = 1–2; <40 days vs. ≥40 days);

*Tj* = calf housing type effect (i = 1–3; slatted floor, floor standing, other pen types);

*Dj* = herd size effect (j = 1–2; <100 vs. ≥100 dairy cow);

*eijk* = residual error.

Through the Multiple Correspondence Analysis (MCA) the relationships among variables (air quality, space available/calf, hygiene of pen walls, average THI, bedding DM, calf health scores, calf mortality) were investigated.

## 3. Results

### 3.1. Calf Management Practices and Mortality

The main information on calf management practices, together with mortality during the preweaning period, obtained from the interviews and measurements are reported in Table 1.

Farms involved in the study differed in herd size: on average, 250 calves born per year were registered, but with a high standard deviation.

Calves were separated from their dam by 24 h from birth but on average they stay with the cow less than 4 h. Weaning time was highly variable among the farms, from a minimum of 40 days to a maximum of 160 days. The milk fed included: whole milk from the farm (32%), milk replacer (32%) or both (36%). It was administered on average 2 times a day mainly by bucket (44%) or bottle in the first days and bucket subsequently (28%).

On the average, calf pen area was 2.35 m^2^/head and calves were kept in individual pens for about 50 days and then moved to group pens. The interval between complete bedding renewal in the calf pens almost corresponds to the time spent by the calf in the individual pen. Calf pens were bedded with straw in all the surveyed farms. Straw bedding Dry Matter (DM) was, on average, 65.6% (SD = 12.0), from 47.2% to 92.2%.

Mortality in the first 24 h was lower than that of 24 h-weaning, namely 2.50% and 4.24%, respectively.

### 3.2. Pen Air Quality, Hygienic Conditions and Calf Health

Air quality, hygiene of calf housing and equipment and calf health conditions are summarized in Table 2.

Microbial counts and ammonia concentrations were determined on air samples from 85 pens regardless of the age of calves. The repeated samples taken over the course of five weeks in 3 individual pens from the day they were empty until five weeks later, with the presence of the calf showed that the permanence of the calf in the cage had only a slight not significant effect on air ammonia concentrations. Moreover, no effect on the microbiological quality of the air was revealed. Considering the different groups of microorganisms in the pen air, a high presence of yeasts and moulds can be noticed. The hygienic conditions of buckets (*n* = 34), teats and esophageal probes (*n* = 31) and pen walls (*n* = 43) were very poor as log_10_ values were always over 2. However, the minimum values found can be considered as clean surfaces. Health condition scores were attributed to a total of 230 calves. Overall, health conditions were good, with a high frequency of zero scores for nasal discharge (≥76%) ocular discharge/ear position (≥72%), cough (≥98%). Only faecal consistency showed a moderate frequency of non-zero scores (almost 40%).

### 3.3. Microclimatic Conditions

Microclimatic conditions in the pens are reported in Table 3.

The highest average ambient temperature (AT) was detected during the afternoon; the average AT of morning and evening were similar, while the lowest AT was detected during the night. On the contrary, during the night the highest value of average relative humidity (RH) was registered, while the lowest value was detected during the afternoon. The highest average light intensity was recorded during the morning (246 lux; SD = 265) and the afternoon (267 lux; SD = 253 lux). There was a wide variability of light intensity with minimum values registered in the morning and in the afternoon of 3.90 and 3.92 lux, respectively, which was very close to nighttime values.

### 3.4. Correlations among Air Quality and Microclimatic Conditions

In Table 4, Pearson’s correlation coefficients among the main variables related to air quality and microclimatic conditions in the calf pens are shown.

High positive correlations were found between average THI in all parts of the day and air SPC and between night maximum THI and SPC. Airborne SPC was also highly positively related to both yeast and mould counts and to ammonia concentration.

### 3.5. Effect of Pen Position, Pen Design and Management on Air Quality, Contamination of Calf Housing and Calf Health

The effect of the position of individual pen (central or external, with reference to the position in the line of calf pens next to each other) was evaluated. In general, air in the pens located in the external position showed more microbial contamination. In particular, the calf pen position significantly affected Enterobacteriaceae count in the pen air (*p* = 0.04).

The evaluation of the effect of pen design on air quality, contamination of pen walls, humidity of bedding material and calf health scores showed that floor standing pens had more contaminated air in terms of yeasts compared to “other pen types” (hutches, group pens; *p* = 0.02).

Frequency of bedding renewal influenced air microbial contamination (Table 5), as well as a calf’s cough. In particular, when the interval of bedding renewal was higher than 40 days, a worsening of the microbiological quality of the air was observed, particularly in terms of the concentrations of *E. coli* (*p* = 0.03), Enterobacteriaceae (*p* = 0.08) and yeasts (*p* = 0.02), as well as an increased frequency of non-zero cough scores (*p* = 0.02) (Figure 1).

### 3.6. Association between Pen Contamination, Microclimatic Conditions, Air Quality and Calf Health

A multiple correspondence analysis (MCA) was performed to extract meaningful information on the relationships among the variables (Figure 2).

One dimension explains 30.0% of the total variation while the second dimension explains 26%. From MCA emerged that, in conditions of high bedding dry matter (dry bedding), both scores for faecal consistency and eye discharge/ear position were low and total calf mortality lower than 6%. These conditions were related to good air quality (low SPC) and low THI (below the average value recorded).

## 4. Discussion

### 4.1. Calf Management Practices and Mortality

The Council Directive 2008/119/EC [16], that lays down minimum standards for the protection of calves, only provides general guidelines concerning calf management practices. This explains the wide variability observed in the present study for calf management practices adopted in the 25 studied farms.

With respect to cow-calf separation, other authors [1] in a survey carried out almost 10 years ago in intensive dairy cattle farms from northern Italy, found a time birth-dam separation about half of that observed in these 25 farms (1.7 vs. 3.7 h). Council Directive 2008/119/EC is completely silent about this issue and within the international literature, different indications are given. Early separation, indeed, is usually considered as the best practice, not only for economic reasons, but also for protecting calf health and minimizing separation distress [17,18]. However, due to the raising concerns of public and consumers about the welfare of farmed animals, longer cow–calf contact is currently being considered as an alternative, since the research evidence does not always support the recommendation for early dairy cow-calf separation, on the basis of calf or cow health [18].

Regarding the administration of colostrum, on average interviewed farmers declared to apply the minimum criteria but it would be advisable they will pay more attention to the timing and quality of the colostrum. The Council Directive provides the indication that each calf must receive bovine colostrum as soon as possible after birth and, in any case, within the first six hours of life to obtain an effective transfer of passive immunity. In the present study, the average time of first colostrum feeding was just below six hours from birth, a worse figure than that obtained in other surveys [1]. Early consumption of an adequate amount of high-quality colostrum is important for the acquisition of passive immunity, which, in turn, may influence predisposition to diseases [19]. The average IgG concentration of colostrum in the present study was just above the threshold of 50 g/L identified for high quality colostrum [20], with a very wide variability, and just a half of the farmers said to regularly measure the quality of the colostrum before administration.

In the present study, all calves were fed on a conventional limit-feeding program (a total of 6 L/head distributed in 2 meals per day) and on average they were weaned before 3 months of age. Both results are very similar to those reported by other authors [1]. Concerning calf milk feeding, the pre-weaning period is critical for calf health and growth, and intensive milk feeding programs may assist post-natal development by improving body growth and organ maturation. In particular, a great deal of research indicated that enhanced milk feeding programs during the first weeks of life can promote body and skeletal growth and calf wellbeing compared with conventional limit-feeding programs [21,22,23]. However, a concern regarding feeding calves with large amounts of milk or milk replacer is that the greater daily gain of the pre-weaning period and the greater body weight at weaning are generally lost by 4 months of age, compared with a limit-feeding programs [24,25].

On the average, calves were disbudded at about 3 weeks of age, but a wide variability was recorded among the studied farms (from 5 to 50 days of life; this is not in line with the Italian regulation [26], which prescribes disbudding by the third week of calf life. Nevertheless, some studies reported increased pain sensitivity when painful procedures such as disbudding are performed at an early age [27].

In the present survey, the average time spent by calves in individual pens was slightly below the limit of 8 weeks imposed by the Council Directive, but some farms left the calves in individual cage well beyond the mentioned period. In comparison with the results of other authors [1], the average period of isolation in our survey was longer of more than 10 days. According to the international literature [28], calves reared individually show low learning abilities, reduced social skills and have difficulty in coping with new situations, revealing poor resilience to stress. In addition, in response to the “End the Cage Age” petition, the EU Commission has pledged to prepare, by the end of 2023, a proposal to phase out the use of individual cages for calves even under 8 weeks of age by 2027 [29]. This will force to study new solutions for housing and managing calves in the first week of life.

As reported by others [4], dairy calves usually show a clear preference for dry sawdust bedding and aversion to concrete lying surfaces, indicating that access to soft and dry bedding is important for welfare of growing calves. The average DM content of bedding material (always straw) in the surveyed farms showed rather good values, more favourable than that reported by other authors [30]. Bedding renewal almost coincided with the average time spent by the calves in the individual pens which suggests that the bedding material in most cases is renewed only at the end of the cycle, although it is added to every few days.

Regarding calf mortality, the overall percentage, including both perinatal mortality in the first 24 h of life and mortality between 24 h and weaning was lower than that obtained from a previous survey carried out in intensive Italian dairy farms [1] but almost double than that reported by other authors [31] for dairy calves in Great Britain (3.87%).

### 4.2. Air Quality, Contamination of Pens and Equipment, Microclimatic Conditions and Calf Health

To the best of our knowledge, few studies exist on air quality in calf housing and Council Directive 2008/119/EC does not provide thresholds for microbial contamination in the pen air. The document only states that insulation, heating and ventilation of the buildings must ensure that air circulation, dust level, temperature, relative air humidity and noxious gas concentrations need to be kept within limits which are not harmful to the calves. Proper ventilation, allowing the availability of fresh and clean air and the removal of dusts, gases, pollutants, moisture and microbes helps to optimise calf health [11,29,32].

The average air bacterial count in the present study was slightly lower than values reported by other authors [30] but higher than the thresholds recommended by others [8] to maintain a low risk of lung problems. Considering that the repeated samples taken over the course of five weeks in 3 individual pens showed that the permanence of the calf in the cage had no significant effect on the microbiological quality of the air, the results of air microbial contamination can be assumed as reliable although the air samples were taken regardless of the time spent by calves in the tested pens.

Air ammonia concentration in the calf pen is related to the risk of calf pneumonia: according to [10], ammonia concentration above 4 ppm was associated with lung consolidation and was also positively related to epithelial cell recovering from broncho-alveolar lavage fluid. However, the results of other studies [11] do not seem to confirm the relationship between high ammonia levels in the pen air and the risk of developing respiratory diseases in calves. Other authors [33,34] indicated ammonia concentration higher than 10 ppm and 25 ppm, respectively, as risk factor for calf health. The results of the present study showed concentrations of ammonia in the pen air much lower than the thresholds suggested, even considering the maximum values.

Council Directive 2008/119/EC, concerning the routine cleaning of calf housing, states that housing, pens, equipment and tools used for calves must be properly cleaned and disinfected to prevent cross-infection and the build-up of disease-carrying microorganisms. In addition, it is specified that feeding and watering equipment must be designed, constructed, placed and maintained so that contamination of the calves’ feed and water is minimised. However, the results of the swab tests revealed that milk and colostrum feeding equipment (buckets, teats and esophageal probes), as well as pen walls were, on average, dirty. In fact, according to other authors [35], a surface can be considered clean with values less than 1 log_10_ Relative Luminescence Units (RLU), not properly cleaned with values between 1.04 and 1.46 log_10_ RLU, dirty with measurements greater than 1.48 log_10_ RLU. According to [36] the sanitization of buckets, teats and others feeding tools through the use of detergents must be able to remove the biofilm created by milk fat, protein and lactose, that may represent a favourable environment for bacterial growth. The hygienic conditions of feeding equipment are of particular concern for colostrum feeding as colostrum contaminated with bacteria can lead to decreased absorption of immunoglobulins due to competition at the level of intestinal epithelium of the calf [37]. Despite the poor hygienic conditions of feeding equipment and pen walls, calves of the studied farms seemed not to show symptoms of poor health conditions according to the results obtained from the evaluation protocol adopted which showed a high percentage of zero scores.

The average temperature detected inside the calf pens in the 25 farms ranged from 7.89 °C in the night to 13.7 °C in the afternoon. Most of calf pens were outdoor (64%) whereas the other ones were indoor or at least under a shelter. As reported in other studies [33,38], a large variation of temperature in the calf pen (as difference between the highest and the lowest temperature values) seems to be associated with high calf mortality. Some authors [39,40] reported as lower critical temperature, at which additional energy is needed for heat production by the animal, the range of 10–15 °C for calves in the first two weeks of life. The critical threshold declines with age to approximately 6–10 °C in older calves. Both values are higher than the minimum temperature detected in the studied farms which could suggest a mild cold stress condition, especially in younger calves. The average RH in the studied farms was similar to that reported by other authors [33], who found out an overall mean humidity in calf pens of 62%. According to [40], relative humidity should be kept below 75% to ensure adequate air hygiene by providing good ventilation and maintaining dry bedding. With regards to THI, some authors [41] inferred that welfare of young calves may be compromised above a THI of 78 and that calves experience significant heat stress above a THI of 88. The maximum THI recorded in the studied farms never exceeded the critical threshold, considering that the study was carried out during the winter season. As for several aspects concerning the minimum standards for the protection of calves, about microclimatic conditions, the Council Directive 2008/119/EC only provides general guidelines. In terms of light intensity, it is only specified that calves must not be kept permanently in darkness and that, in case of artificial lighting, it must be provided for a period at least equivalent to the period of natural daylight. The international literature [42] reported a recommended light intensity of 100–150 lux daytime and 5 lux at nighttime, while other authors [43] reported for Europe the recommendations for minimum light intensities in cattle barns in a range from 20 to 120 lux. On average, in the studied farms an appropriate lighting was provided although very low minimum values were registered even during the daytime, especially in the indoor pens.

A high correlation coefficient was found for airborne SPC and mould concentration; also, average THI and maximum night THI were highly correlated with air SPC in the pen air. It is very well known, indeed, that, in any environment, humidity is positively associated with concentration of bacteria and other microorganisms [44,45]. A good livestock housing design and maintenance is a fundamental factor for avoiding the proliferation of microorganisms in the environment, and an essential aspect to be considered is humidity [3]. As stated by others [46], the level of humidity is important for the welfare of animals because the infectivity of pathogens found in the environment depends on this level. Likewise, the bacterial concentration of the air is affected by temperature, as if the temperature moves away from the optimal range, the relative humidity will be modified, to limit the proliferation of microorganisms and moulds [3].

Results obtained showed that pen design and position affect air microbial contamination: air of “floor standing pens” was richer in yeasts compared with “other pen types”. In addition, pens located in the external position of the pen line showed more contaminated air, in terms of Enterobacteriaceae than internal ones. This latter result is unexpected as pens in the internal position in the pen line, being less ventilated, were supposed to have more contaminated air. However, it could be that the external pens were more subjected to the transfer of bacteria from the adjacent parts of the barn, in particular from the barn for adult cows, the calving area or the pens devoted to sick animals. This outcome can suggest the importance of keeping calf pens as separate as possible from the rest of the barn and the adult animals, to reduce air contamination. Other authors previously found that in the herds where calf pens were placed along an outer wall of the adult barn, calves had an increased risk of suffering from diarrhoea and tended to have an increased risk of other infectious diseases [11].

It is worth noting that *E. coli*, Enterobacteriaceae and yeast counts in the air of calf pens were higher when the renewal of bedding material was rarely performed. At the same time, long intervals between bedding renewals were associated to higher cough scores in calves. This result suggests the importance of a proper bedding management, to favour dry environment and clean air for calves. Regarding yeast, some authors [6] stated that they may be allergenic to certain individuals when present in sufficient concentrations and can affect both animal and human health. Recent studies highlighted that some farm practices can have significant effects on the type of microorganisms in the sheep barn and revealed positive correlations between the concentrations of lactic bacteria and yeast in the barn air and their concentrations in the milk [47,48]. These studies underline the importance of deepening our knowledge about the microbiological quality of the barn air.

Results obtained by MCA showed that bedding dry matter, microclimatic conditions and microbiological air quality can be related to calf health scores and mortality. In particular, low mortality rates were associated with low faecal and eye/ear scores, good hygienic condition of the pen walls, dry bedding, low THI and low SPC. Any management practice that can enhance bedding DM content as frequent renewal, proper ventilation and regular addition of dry bedding material can be favourable for improving air quality and contributing to maintain good health conditions of the calves.

The study has some limitations that should be addressed in future research. In the present study each farm was visited once, and the air samples were taken in the selected pens regardless of the age of the calves in the pens. To test the possible effect of this bias, in one farm air quality was monitored over the course of five weeks in 3 individual pens to observe possible changes over time. The result of the test showed that the length of the stay of the calf had no significant effect on the microbiological quality of the air in the pen, but in the future this possible source of bias has to be taken into account especially in the case of measurements during the summer season. Another limitation is represented by the season chosen for the measurements: the winter season probably limited the release of ammonia and the microbial growth. During the summer the relationships among microclimatic variables and airborne bacteria and fungi may change significantly suggesting the need for further research to overcome the limitations of the present study.

## 5. Conclusions

In conclusion, the study revealed that microbiological air quality in the calf pens was affected by Temperature Humidity Index in the pen. In addition, pen design and position had an influence on air quality: in particular floor standing pens showed high yeast contamination. Counts of *E. coli*, Enterobacteriaceae and yeasts in the pen air were higher and calf cough increased as the renewal interval of bedding material became longer. Maintaining bedding humidity low and controlling microclimatic conditions, such as relative humidity and THI, can enhance air microbiological quality in the pen and seem to be related to decreased calf diseases and mortality rate. The study of the microbial quality of air in the barn can help to identify best practices to ensure a healthy housing environment. However, at the moment, our knowledge about the microbial species present in the air of livestock housing, their distribution and the main factors influencing them are still very limited. Although calf management practices may differ between regions and countries, some of the results of this study might be generalizable to intensive dairy farming systems from other countries.

Information on pre-weaning calf management and mortality rates in the studied farms showed that there is still room for improvement of management practices in intensive dairy cattle farming in northern Italy. In particular, hygienic conditions of the milk and colostrum feeding equipment used for calves emerged as a hot spot that can pose a hazard for calf health and deserves to be further explored.

## Figures and Tables

**Figure 1 animals-12-02286-f001:**
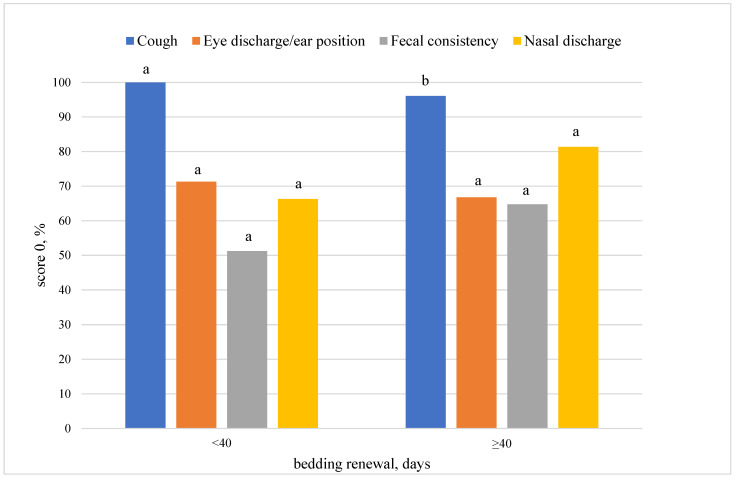
Effect of frequency of bedding renewal on calf health scores. Score 0 corresponds to normal condition. Different letters between bars of the same colour mean *p* < 0.05.

**Figure 2 animals-12-02286-f002:**
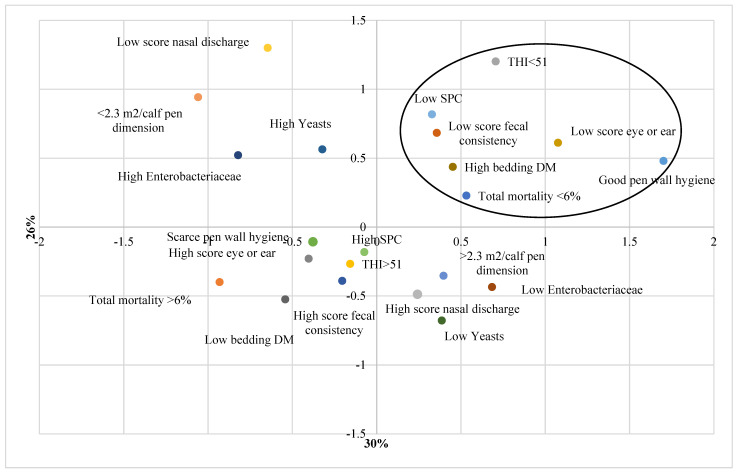
Multiple Correspondence Analysis highlighting the relationships among microclimate variables, pen area, pen hygiene, air microbial contamination, calf health scores and mortality. SPC = Standard Plate Count. THI = Temperature Humidity Index. DM = Dry Matter. SPC = Standard Plate Count.

**Table 1 animals-12-02286-t001:** Calf management practices and mortality during the preweaning period (*n* = 25).

Variable	Unit	Mean	SD	Minimum	Maximum
Calves born/farm	no./year	250	244	55.0	1080
Time birth-dam separation	h	3.69	6.94	0	24.0
Time birth-1st colostrum meal	h	5.64	2.69	1.50	12.0
Colostrum IgG	g/L	52.8	23.8	4.0	158
Time birth-1st water offer	d	8.64	9.07	0	30.0
Time birth-1st starter meal	d	12.5	18.8	0	90.0
Time birth-disbudding	d	20.6	11.2	5.0	50.0
Time birth-1st hay offer	d	49.4	44.8	2.5	180
Individual milk feeding	L/meal	3.00	0.96	2.00	6.00
Individual starter feeding	kg/day	1.58	0.95	0.65	3.50
Weaning time	d	81.1	25.7	40.0	160
Time spent in individual pen	d	51.2	27.1	7.5	120
Frequency of bedding renewal	d	41.4	35.5	4.5	120
Calf pen area	m^2^/head	2.35	1.55	1.25	8.00
Mortality birth-24 h	%	2.50	3.83	0	18.0
Mortality 24 h-weaning	%	4.24	2.34	0	9.46

**Table 2 animals-12-02286-t002:** Air quality, contamination of pen walls and feeding equipment, and calf health scores (*n* = 25).

Item	Unit	Mean	SD	Minimum	Maximum
Air quality					
*Escherichia coli*	log_10_ MPN ^1^/m^3^	1.20	1.11	0	3.80
Enterobacteriaceae	log_10_ MPN ^1^/m^3^	2.41	0.96	0	4.03
SPC ^2^	log_10_ MPN ^1^/m^3^	4.51	0.52	3.37	5.12
Yeasts	log_10_ MPN ^1^/m^3^	3.32	1.05	0.00	5.12
Moulds	log_10_ MPN ^1^/m^3^	3.82	0.63	2.84	5.12
Ammonia concentration	ppm	0.66	0.53	0.07	2.00
Contamination of feeding equipment and pen walls					
Buckets	log_10_ RLU ^3^	2.27	0.69	0.72	3.79
Teats	log_10_ RLU ^3^	2.53	0.94	0.77	3.98
Pen walls	log_10_ RLU ^3^	2.16	0.55	0.80	3.13
Calf health scores					
Nasal discharge	% score 0	75.7	19.6	25.0	100
Eye discharge/ear position	% score 0	72.0	17.1	25.0	100
Cough	% score 0	97.9	6.4	71.4	100
Fecal consistency	% score 0	60.6	24.9	0	100

^1^ MPN = Most Probable Number. ^2^ SPC = Standard Plate Count. ^3^ RLU= Relative Luminescence Unit.

**Table 3 animals-12-02286-t003:** Microclimatic conditions in calf pens (*n* = 25).

Item	Unit	Mean	SD	Minimum	Maximum
Night					
Temperature	°C	7.89	4.11	2.48	19.3
RH ^1^	%	72.0	12.2	49.3	95.7
THI ^2^		47.4	6.66	37.1	64.8
Maximum temperature	°C	16.2	4.85	9.49	24.0
Maximum RH ^1^	%	91.5	6.92	74.7	100
Maximum THI ^2^		59.5	6.70	49.9	69.6
Morning					
Temperature	°C	10.2	3.73	3.44	20.5
RH ^1^	%	67.5	13.8	46.5	93.2
THI ^2^		50.9	6.3	38.8	67.9
Maximum temperature	°C	21.3	2.7	15.0	28.3
Maximum RH ^1^	%	93.1	6.09	74.3	100
Maximum THI ^2^		66.4	3.27	58.8	73.7
Afternoon					
Temperature	°C	13.7	4.22	5.27	19.4
RH ^1^	%	53.2	18.6	29.4	86.0
THI ^2^		56.2	6.42	42.9	63.6
Maximum temperature	°C	23.3	4.8	9.6	30.0
Maximum RH ^1^	%	81.2	16.5	39.5	100
Maximum THI ^2^		68.2	5.8	50.1	75.9
Evening					
Temperature	°C	10.1	3.71	3.36	16.2
RH ^1^	%	64.8	15.2	41.9	93.5
THI ^2^		51.0	6.04	38.6	59.9
Maximum temperature	°C	22.1	4.17	11.4	29.4
Maximum RH ^1^	%	87.7	9.26	64.0	100
Maximum THI^2^		67.0	5.46	52.9	79.2

^1^ RH = Relative Humidity. ^2^ THI = Temperature Humidity Index.

**Table 4 animals-12-02286-t004:** Pearson’s correlation coefficients among variables related to air quality and microclimatic conditions in the calf pens.

	*E. coli*	Enterobacteriaceae	SPC ^1^	Yeasts	Moulds	Night THI ^2^	Night Max THI^2^	Morning THI ^2^	Morning Max THI ^2^	Afternoon THI ^2^	Afternoon Max THI ^2^	Evening THI ^2^	Evening Max THI ^2^	Ammonia
*E. coli*	1.0													
Enterobacteriaceae	**0.67**	1.00												
SPC	−0.02	0.20	1.00											
Yeast	0.12	0.28	**0.47**	1.00										
Moulds	−0.05	0.36	**0.71**	**0.54**	1.00									
Night THI ^2^	0.32	0.43	**0.63**	0.24	0.29	1.00								
Night max THI ^2^	0.38	**0.55**	**0.61**	0.43	**0.46**	**0.80**	1.00							
Morning THI ^2^	−0.04	0.08	**0.68**	0.22	0.32	**0.85**	**0.65**	1.00						
Morning max THI ^2^	0.23	−0.18	0.22	0.05	−0.04	0.43	0.26	**0.56**	1.00					
Afternoon THI ^2^	−0.33	−0.35	**0.52**	0.15	0.34	**0.44**	0.39	**0.79**	**0.48**	1.00				
Afternoon max THI ^2^	**−0.54**	−0.44	0.04	−0.09	0.00	−0.03	−0.07	0.29	0.37	**0.64**	1.00			
Evening THI ^2^	−0.07	0.06	**0.71**	0.31	0.41	**0.77**	**0.63**	**0.90**	**0.48**	**0.85**	**0.47**	1.00		
Evening max THI ^2^	−0.23	−0.08	0.27	−0.15	0.30	0.17	0.07	0.30	0.22	0.34	**0.47**	0.39	1.00	
Ammonia	−0.09	0.32	**0.50**	0.13	0.14	0.37	0.33	0.38	−0.09	0.47	0.18	**0.53**	0.06	1.00

Bold font for *p* < 0.05. ^1^ SPC = Standard Plate Count. **^2^** THI = Temperature Humidity Index.

**Table 5 animals-12-02286-t005:** Effect of interval between bedding renewals on air quality, contamination of pen walls and humidity of bedding material.

Item	Unit	<40 d Bedding Renewal	≥40 d Bedding Renewal	SE	*P*
		*n* = 15	*n* = 10		
		Air quality			
*Escherichia coli*	log_10_ MPN ^1^/m^3^	0.41	1.69	0.5	0.03
Enterobacteriaceae	log_10_ MPN ^1^/m^3^	2.11	3.25	0.7	0.08
Yeasts	log_10_ MPN ^1^/m^3^	2.04	3.05	0.37	0.02
Moulds	log_10_ MPN ^1^/m^3^	3.72	4.07	0.31	0.30
SPC ^2^	log_10_ MPN ^1^/m^3^	4.28	4.61	0.27	0.26
Ammonia concentration	ppm	0.59	0.35	0.27	0.41
Pen wall contamination					
Pen wall hygiene	log_10_ RLU ^3^	2.42	2.41	0.26	0.96
		Bedding dry matter			
Bedding DM ^4^	%	70.1	73.3	5.97	0.63

^1^ MPN = Most Probable Number. ^2^ SPC = Standard Plate Count. ^3^ RLU = Relative Luminescence Unit. ^4^ DM = Dry Matter.

## Data Availability

The data supporting the reported results are available on request from the corresponding author.

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
