# Peer review of "Air Quality, Management Practices and Calf Health in Italian Dairy Cattle Farms"

_animals, 2022, doi:10.3390/ani12172286_

Round 1

Reviewer 1 Report

General comments:

The authors report interesting results on the air quality and the management practices at farm level. There is limited information about the airborne bacteria and their effects in the farms, thus the topic of the manuscript is interesting. Results are relevant and can be of interest for farmers, veterinarians, and consumers, and set the grounds for further analytical surveillance programs in dairy industry. However, some points of the manuscript require attention. It would be advisable to study differences between winter and summer, since specific studies in small dairy ruminants have revealed differences in the environmental microbiology of farms. In addition, some tables are confusing, they should have subsections to clarify the results. Specific comments are below.

Specific comments:

  • Line 110: which was the final volume? How many minutes was the sampler working?
  • Air quality evaluation: What was the maximum distance between farms and the laboratory? Were the microbiological samples incubated on the same day? It is important to note that airborne samples are very labile and must be analyzed and cultured in a short time. 
  • Line 141: “Results >10 RLU is considered positive” What are you based on? It is necessary to complete the paragraph with some references.
  • Line 177: “Statistical analysis” should be paragraph 2.5
  • Results: To improve the discussion, there are recent studies of environmental microbiology in small dairy ruminants in Spain that should be compared in this study, since they study the correlations between different airborne microorganisms such as yeasts and lactic acid bacteria with the management of farm and air quality.

Author Response

General comments:

The authors report interesting results on the air quality and the management practices at farm level. There is limited information about the airborne bacteria and their effects in the farms, thus the topic of the manuscript is interesting. Results are relevant and can be of interest for farmers, veterinarians, and consumers, and set the grounds for further analytical surveillance programs in dairy industry. However, some points of the manuscript require attention. It would be advisable to study differences between winter and summer, since specific studies in small dairy ruminants have revealed differences in the environmental microbiology of farms. In addition, some tables are confusing, they should have subsections to clarify the results. Specific comments are below.

Au: We fully agree with your comment on the effect of season. As soon as possible, we intend to plan a study to evaluate the effect of season on the air microbiology in the calf housing and on the other variables. However, in the present study, we preferred to have similar climatic conditions in order to highlight the differences among farms characterized by different calf management and housing systems.

We added subsections to the tables, as suggested.

Specific comments:

  • Line 110: which was the final volume? How many minutes was the sampler working?

Au: We rephrased the sentence, in order to better explain air sampling procedure.

  • Air quality evaluation: What was the maximum distance between farms and the laboratory? Were the microbiological samples incubated on the same day? It is important to note that airborne samples are very labile and must be analyzed and cultured in a short time. 

Au: The farthest farm was less than 100 km from the laboratory and all the Petri dishes were incubated on the same day of collection. In addition, the samples were all managed and stored very carefully, before being incubated for preventing any alteration. We added more details in the Material and methods section

  • Line 141: “Results >10 RLU is considered positive” What are you based on? It is necessary to complete the paragraph with some references.

Au: You are right, we clarified the point. In the discussion section a reference is present

  • Line 177: “Statistical analysis” should be paragraph 2.5

Au: We added the number of the paragraph

  • Results: To improve the discussion, there are recent studies of environmental microbiology in small dairy ruminants in Spain that should be compared in this study, since they study the correlations between different airborne microorganisms such as yeasts and lactic acid bacteria with the management of farm and air quality.

Au: Thank you for your suggestion. In the discussion section we mentioned two studies on air microbial quality in sheep farms

Reviewer 2 Report

The manuscript entitled:" Air quality, management practices and dairy calf health" submitted by Bonizzi et al. to Animals discussed the air microbiological contamination and ammonia concentration in the calf pens, the environmental and management factors affecting air quality, and the relationship with calf health and mortality. Also, the calf management practices adopted in intensive dairy farming in northern Italy were represented in this study. The overall merit of this manuscript is high as the authors detailed calf management practices, mortality, air quality, as well as hygiene of pens, equipment, microclimatic conditions, and calf health by evidencing their contribution to calf health conditions and mortality rates via microbial contamination, which may hazardous the intensive dairy farms.

However, I have minor questions, comments, and suggestions, hoping they may help improve the manuscript, which is as follows:-

1-    I think the title is not reflecting the comprehensive content of the manuscript considering the country of the study and the nature of the research; in this context, I suggest adding words like survey, microbial, and north Italy farms.

2-    In key words, I believe dairy farm management and calf health would be much more comprehensive than dairy calf and calf management.

3-    In the abstract, it will be better to mention the study's time frame (Ex. January to March 2020 and from December to March 2021).

4-    Introduction reflected the topic very well and helped create a general picture about the topic, but if the authors would give some examples of air-borne diseases that could be attributed to pens crowdedness and a non-hygienic environment like the bovine respiratory disease (BRD) will be of much interest for readers.

5-    The rest of the manuscript is well written and regularly prepared, analyzed, and designed; however, a minor language proofreading is needed to enhance the content; also, in conclusion, why the authors did not write some recommendations for further research to surpass the gabs on this study and tackle dairy farms related management problems.

Author Response

The manuscript entitled:" Air quality, management practices and dairy calf health" submitted by Bonizzi et al. to Animals discussed the air microbiological contamination and ammonia concentration in the calf pens, the environmental and management factors affecting air quality, and the relationship with calf health and mortality. Also, the calf management practices adopted in intensive dairy farming in northern Italy were represented in this study. The overall merit of this manuscript is high as the authors detailed calf management practices, mortality, air quality, as well as hygiene of pens, equipment, microclimatic conditions, and calf health by evidencing their contribution to calf health conditions and mortality rates via microbial contamination, which may hazardous the intensive dairy farms.

However, I have minor questions, comments, and suggestions, hoping they may help improve the manuscript, which is as follows:-

  • I think the title is not reflecting the comprehensive content of the manuscript considering the country of the study and the nature of the research; in this context, I suggest adding words like survey, microbial, and north Italy farms.

Au: Thank you for your suggestions. We modified the title adding the specification about the country. However, we prefer to avoid the term “survey” because it can suggest mainly the collection of data through interviews rather than measurements. Moreover, we prefer not to use the term “microbial”, because we want to highlight also the ammonia air concentration data that we provided in the present study, since in the international literature there are only few studies about ammonia air concentration in calf pens.

2-    In key words, I believe dairy farm management and calf health would be much more comprehensive than dairy calf and calf management.

Au: We modified the key words according to the suggestion.

3-    In the abstract, it will be better to mention the study's time frame (Ex. January to March 2020 and from December to March 2021).

Au: We followed your suggestion

4-    Introduction reflected the topic very well and helped create a general picture about the topic, but if the authors would give some examples of air-borne diseases that could be attributed to pens crowdedness and a non-hygienic environment like the bovine respiratory disease (BRD) will be of much interest for readers.

Au: We added some examples and references to the introduction

5-    The rest of the manuscript is well written and regularly prepared, analyzed, and designed; however, a minor language proofreading is needed to enhance the content; also, in conclusion, why the authors did not write some recommendations for further research to surpass the gabs on this study and tackle dairy farms related management problems.

       Au: Both in the last part of the discussion and in the conclusion section there were still some sentences with recommendations for further research. We reinforced them. An extensive revision of language was performed